# Relationship between Difficulty in Emotion Regulation and Internet Addiction in College Students: A One-Year Prospective Study

**DOI:** 10.3390/ijerph17134766

**Published:** 2020-07-02

**Authors:** Jui-Kang Tsai, Wei-Hsin Lu, Ray C. Hsiao, Huei-Fan Hu, Cheng-Fang Yen

**Affiliations:** 1Department of Psychiatry, Kaohsiung Armed Forces General Hospital, Kaohsiung 80284, Taiwan; regon720@gmail.com; 2Department of Nursing, Meiho University, Pingtung 91202, Taiwan; 3Department of Psychiatry, Ditmanson Medical Foundation Chia-Yi Christian Hospital, Chia-Yi City 60002, Taiwan; wiiseen@gmail.com; 4Department of Senior Citizen Service Management, Chia Nan University of Pharmacy and Science, Tainan 71710, Taiwan; 5Department of Psychiatry and Behavioral Sciences, University of Washington School of Medicine, Seattle, WA 98105, USA; rhsiao@u.washington.edu; 6Department of Psychiatry, Children’s Hospital and Regional Medical Center, Seattle, WA 98105, USA; 7Department of Psychiatry, Tainan Municipal Hospital (managed by Show Chwan Medical Care Corporation), Tainan 70173, Taiwan; 8Department of Psychiatry, School of Medicine and Graduate Institute of Medicine, College of Medicine, Kaohsiung Medical University, Kaohsiung 80708, Taiwan; 9Department of Psychiatry, Kaohsiung Medical University Hospital, Kaohsiung 80708, Taiwan

**Keywords:** internet addiction, emotion regulation, college student

## Abstract

This prospective study evaluated the predictive effect of difficulty in emotion regulation on the occurrence and remission of Internet addiction (IA) and determined whether IA has a role in changing emotion regulation among college students during a follow-up period of 1 year. A total of 500 college students (262 women and 238 men) were recruited. In baseline and follow-up investigations, the levels of IA and difficulty in emotion regulation were evaluated using the Chen Internet Addiction Scale and the Difficulties in Emotion Regulation Scale (DERS), respectively. The results indicated that the subscale of impulse control difficulties on the DERS predicted the incidence of IA during the follow-up period of 1 year in male participants (*t* = −2.875, *p* = 0.005), whereas no subscale on the DERS predicted the remission of IA. IA did not predict the change in difficulties in emotion regulation. The subscale of impulse control difficulties on the DERS predicted the occurrence of IA in the college students and warrants early intervention.

## 1. Introduction

The Internet has become a key resource in society and everyday life. At social and individual levels, the Internet facilitates interpersonal communication, provides entertainment, and helps individuals create new social networks [1,2]. However, inappropriate or excessive Internet use may result in Internet addiction (IA) and negative life outcomes [3]. People who have IA may have difficulties in controlling their Internet usage and develop problems in occupational or academic performance and in social relationships, as well as psychological and physical problems [4]. Although the Diagnostic and Statistical Manual of Mental Disorders, Fifth Edition (DSM-5) does not list IA as a formal psychiatric diagnosis, it calls for research to examine Internet gaming disorder, a common type of IA, in terms of its course, symptoms, and underlying mechanisms [5]. Therefore, IA is an issue that warrants further study.

A multinational meta-analysis provided estimates of the global prevalence of IA, which varies across world regions, with the highest prevalence of 10.9% in the Middle East and the lowest prevalence of 2.6% in Northern and Western Europe [6]. The prevalence rate of IA among college students is 3.7% in Japan [7], 8–13% in Taiwan [8], and 13.6% in China [9]. College students may use the Internet for studying, gaming, social networking, gambling, chatting, shopping, and watching pornographic videos [10,11]. However, given that college students have free and unlimited access to the Internet, have flexible schedules, and are free from their parents’ interference, they were identified as having a high risk for IA [12].

There have many studies examining the relationship between IA and personality characteristics such as neuroticism [13,14,15], conscientiousness [13,14,15], agreeableness [13,14,15], boredom proneness [16], borderline personality characteristics [17], reinforcement sensitivity [18], and frustration intolerance [18]. Emotion regulation is the attempt to alter emotional experiences via the initiation, maintenance, or modification of frequency, intensity, or duration of emotional experiences [19]. Difficulties with emotion regulation are believed to be risk factors for addiction [20,21]. Gratz and Roemer proposed that emotion regulation involves multiple conceptions, including (a) awareness and understanding of emotions, (b) acceptance of emotions, (c) ability to control impulsive behaviors and behave in accordance with desired goals when experiencing negative emotions, and (d) ability to use situationally appropriate emotion regulation-n strategies flexibly to modulate emotional responses as desired in order to meet individual goals and situational demands [22]. The relative absence of any or all of these abilities would indicate the presence of difficulties in emotion regulation [22]. 

Research has found that young people with IA had more difficulty in identifying and describing emotions, understanding emotional reactions, and controlling their impulsive behaviors in response to negative emotional experiences, and were less able to use effective emotion regulation strategies than adolescents without IA [23,24,25,26,27]. It is hypothesized that inadequate prefrontal cognitive control to suppress their negative emotions may result in impulsivity and increase the risk of IA [26]. A two-year prospective study also reported that emotional problems, such as depression and social phobia, predict the occurrence of IA in adolescents [28]. However, research also found that Internet use is a common strategy that people use to alleviate emotional distress [29,30,31]. Based on the principle of functional analysis in behavioral psychology [32], many forms of problematic behaviors and psychopathology can be conceptualized as efforts to escape and avoid emotions, thoughts, memories, and other private experiences [33]. Although IA is generally considered an unhealthy effort to cope with stress and negative emotions [29], whether IA deteriorates or improves the function of emotion regulation warrants survey.

No prospective study has examined the bidirectional relationship between IA and difficulties in emotion regulation. The prospective study design can provide the temporal relationships between IA and difficulties in emotion regulation and infer the causal relationship between them. The aims of this prospective study were to evaluate the predictive effect of difficulty in emotion regulation on the occurrence and remission of IA, and to determine whether IA can predict the change in difficulty in emotion regulation among college students during the follow-up period of 1 year. In particular, given that emotion regulation contains multiple dimensions of conceptions [22], this study aimed to examine whether the relationships between IA and emotion regulation vary according to various dimensions of emotion regulation. Based on the results of previous studies described above, we hypothesized that difficulty in emotion regulation may predict the occurrence of IA and the non-remission of IA during the period of one-year follow-up. Moreover, we also hypothesized that IA may worsen emotion regulation during the period of 1 year.

## 2. Materials and Methods

### 2.1. Participants

Participants were recruited using an advertisement posted for college students aged between 20 and 30 years. A total of 500 college students (262 women and 238 men) participated in this study. Regarding the sample size, a previous study found that 8–13% of college students in Taiwan had IA [8]. The sample of 500 participants was determined as adequate based on the estimation with 80% power, 95% confidence interval, and statistically significant level at 5% [34]. The mean age of the participants was 22.1 years (standard deviation (SD): 1.8 years). Informed consent was obtained from all the participants prior to assessment. The study was approved by the Institutional Review Board of Kaohsiung Medical University Hospital (KMUH-IRB-20120249).

### 2.2. Measures

#### 2.2.1. Chen Internet Addiction Scale (CIAS)

We used the self-administered Chen IA Scale (CIAS) to assess the participants’ severity of IA in the month preceding the study [35]. The CIAS contains 26 items rated on a 4-point Likert scale with the minimum and maximum scores of 26 and 104, respectively [35]. A higher total score indicates more severe IA. The internal reliability (Cronbach’s α) of the CIAS in the present study was 0.93. According to diagnostic criteria for IA, a score of 68 out of the total CIAS score has the highest diagnostic accuracy and accepted sensitivity and specificity for IA [36]. Accordingly, those with CIAS scores of 68 or more were classified as those with IA.

#### 2.2.2. Difficulties in Emotion Regulation Scale (DERS)

The DERS is a 36-item self-reported measure developed to assess clinically relevant difficulties in emotion regulation [22]. Items are scored on six scales: Nonacceptance of Emotional Responses (6 items), Difficulties Engaging in Goal-Directed Behavior (5 items), Impulse Control Difficulties (6 items), Lack of Emotional Awareness (6 items), Limited Access to Emotion Regulation Strategies (8 items), and Lack of Emotional Clarity (5 items). Participants are asked to indicate how often each of the 36 items applied to them on a 5-point Likert scale ranging from 1 (almost never) to 5 (almost always). Subscale scores are obtained by summating corresponding items. Cronbach’s α for the Taiwanese version of the DERS ranged from 0.72 to 0.81. Higher scores on the DERS indicate greater difficulties in emotion regulation.

#### 2.2.3. Demographic Characteristics

The demographic characteristics of participants collected in this study were sex (female or male) and age (years).

### 2.3. Study Process and Statistical Analysis

In the initial assessment (Stage 1), all the participants were invited to complete the CIAS and DERS. One year later (Stage 2), the participants were invited to complete the CIAS and DERS again. According to the scores of the CIAS at Stage 1 and Stage 2, the participants were classified into one of four groups (Figure 1). The participants deemed not to be addicted at Stage 1 were stratified into the Persistent no IA (PNIA) and New IA (NIA) groups based on subsequent non-addiction and addiction statuses at Stage 2, respectively. The remaining participants, who were initially deemed to be addicted to the Internet at Stage 1, were stratified into the groups of Remitted IA (RIA) and Persistent IA (PIA) based on remission and continuation of the behavior at Stage 2, respectively.

The incidence rate of IA at Stage 2 and its relationship with difficulty in emotion regulation at Stage 1 were examined in the PNIA and NIA groups. The remission rate of IA at Stage 2 and its relationship with difficulty in emotion regulation at Stage 1 were examined in the RIA and PIA groups. The chi-square test was utilized to evaluate category variables, and Student’s t-test was used for continuous variables.

We evaluated whether changes in the difficulty of emotion regulation during the one-year period were different between the college students with and without IA at Stage 1. A repeated-measures two-way analysis of variance (ANOVA) with investigation stages (Stage 1 vs. Stage 2) as a within-subjects factor and the groups (with or without IA at Stage 1) as a between-subjects factor was performed for DERS scores. All statistical analyses were performed using SPSS version 20.0 (SPSS Inc., Chicago, IL, USA). Because of the existence of multiple comparisons, *p*-values of <0.008 (0.05/6) were considered statistically significant.

## 3. Results

A total of 324 college students (65.8%, 169 women and 155 men) underwent the follow-up assessment 1 year later. Of the 176 participants (93 women and 83 men) who did not participate in follow-up assessment, 96 (54.5%) could not be contacted, 48 (27.3%) refused to participate in follow-up assessment, and 32 (18.2%) had the desire to participate but were unable to do so due to work or army service. The results of comparing demographic data and the levels of IA on the CIAS and difficulties in emotion regulation on the DERS at initial assessment between participants who participated in and who did not participate in follow-up assessment are shown in Table 1. No differences were found in gender, age, and the levels of IA and difficulties in emotion regulation between these two groups (all *p* > 0.008).

Of the 268 participants in the PNIA and NIA groups who had no IA at Stage 1, 20 were deemed to have IA at Stage 2 (the NIA group), resulting in an incidence rate of 7.5% during the one-year period. Of the 56 individuals in the RIA and PIA groups who had IA at Stage 1, 26 were classified as being without IA at Stage 2 (the RIA group), indicating a one-year remission rate of 46.4%.

Comparisons of demographic characteristics and difficulties in emotion regulation between the PNIA and NIA groups are presented in Table 2. The results revealed that compared with the PNIA group, the NIA group had more severe impulse control difficulties on the DERS at Stage 1, revealing that impulse control difficulties at Stage 1 predicted the incidence of IA at Stage 2 during the follow-up period of 1 year. We further stratified the participants by gender and found that the prediction of impulse control difficulties for the incidence of IA existed in only male (*t* = −2.875, *p* = 0.005) but not in female participants (*t* = −1.270, *p* = 0.206).

Comparisons of demographic characteristics and difficulties in emotion regulation at Stage 1 between the RIA and PIA groups are presented in Table 1. No dimension of difficulty in emotion regulation at Stage 1 was significantly associated with the remission of IA at Stage 2.

The results of the repeated-measures ANOVA for the effect of IA at Stage 1 on changes in DERS scores from Stage 1 to Stage 2 are shown in Table 3. Regarding the results of within-subject analysis, scores for the subscales of lack of emotional awareness (*p* = 0.017) and limited access to emotion regulation strategies (*p* = 0.022) tended to decrease more from Stage 1 to Stage 2 in the IA group than in the non-IA group; however, the results did not reach a statistically significant level. Regarding the results of between-subject analysis, there were significant differences in non-acceptance of emotional responses (*p* = 0.006), difficulties engaging in goal-directed behavior (*p* < 0.001), impulse control difficulties (*p* < 0.001), limited access to emotion regulation strategies (*p* < 0.001), and lack of emotional clarity (*p* < 0.001) between the participants with and without IA at Stage 1.

To explore the effects of IA on the changes in difficulties in emotional regulation, the scores on the DERS were further compared between the Stage 1 and Stage 2 investigations using a paired t-test in the participants with and without IA at Stage 1 (Table 4). The results of the paired t-test demonstrated that in the IA group, the scores on the subscale of lack of emotional awareness decreased significantly from Stage 1 to Stage 2. In the non-IA group, the scores on the subscales of lack of emotional awareness and lack of emotional clarity decreased significantly from Stage 1 to Stage 2.

## 4. Discussion

This study revealed that more severe impulse control difficulties predicted a higher incidence of IA during the follow-up period of 1 year in male participants. Related cross-sectional studies have found a significant association between impulsivity and Internet gaming disorder in adolescents and young adults [37,38]. A longitudinal study also indicated that impulsivity is a risk factor for Internet gaming disorders among adolescents [39]. Loss of control of Internet use is an essential criterion for IA [36]; therefore, impulse control difficulties could make individuals more susceptible to the rewarding effects of Internet use and contribute to vulnerability to IA. Impulse control difficulties may also negatively affect young adults’ relationships with others and thus increase their desire to seek out friendship and joy from the Internet. Moreover, a Go/NoGo study reported that college students with IA had lower electrophysiological activation in the conflict detection stage than did those without IA [40], indicating that they had lower efficiency in information processing than did their peers without IA. Low efficiency in information processing may limit individuals’ ability to access effective emotion regulation strategies. Cognitive-behavioral therapy for IA aims to increase the clients’ ability of impulse control by training clients’ skills to monitor their inner feelings and control impulse behaviors [41,42,43]. Research also found that electro-acupuncture had an advantage over psychological intervention in terms of impulsivity control in adolescents with IA [44]. The present study found a gender difference in the prediction of impulse control difficulties for the incidence of IA. Previous studies have also found gender differences in the psychopathology of IA [45,46]. However, the numbers of participants with the incidence of IA during the one-year follow-up period were small. Further study with more participants is warranted to replicate this result.

The one-year remission rate of our study was 46.4%. A previous study on young adolescents in Taiwan found the one-year remission rate of IA was 49.5% [28]. A study on Hong Kong Chinese secondary school students found the one-year remission rate of IA was 59.29/100 person-years [47]. A study on German adolescents found that the one-year remission rate of IA was 71.6% [48]. The results of the present and previous studies indicated that incidence of remission was high without noticeable interventions during adolescence and adulthood. Although the present study did not find a predictive effect of difficulties in emotion regulation on the remission of IA, a previous study found that a lower level of maladaptive emotion regulation strategies significantly predicted for remission one year later [48], indicating that programs enhancing emotion regulation are important for helping people to reduce the severity of IA.

The results of ANOVA examining the changes in difficulties in emotion regulation from Stage 1 to Stage 2 did not show statistically significant differences between the IA and non-IA groups. The results did not support our hypothesis that IA may worsen emotion regulation. The score on the subscale of lack of emotional awareness decreased significantly from Stage 1 to Stage 2 in both IA and non-IA groups, whereas the score on the subscale of lack of emotional clarity decreased significantly in only the non-IA group. However, it is noteworthy that the IA group had higher scores of all subscales of the DERS than the non-IA group at the initial investigation (*p*-values ranging from <0.001 to 0.003). Research has found that people with negative emotional states, such as depression, anxiety, and feelings of loneliness, were more likely to use the Internet to manage their emotional problems [29,30,31]. Internet use might be a way for people with negative emotions to experience pleasure and avoid unpleasant experiences [49]. Research found that maladaptive emotion regulation strategies could predict the maintenance of problematic Internet use, whereas a targeted positive development of emotion regulation in childhood and adolescence could promote spontaneous remission of problematic Internet use [48]. Therefore, mental health professionals need to provide intervention programs for people with IA to improve their ability of emotion regulation.

The present study has several limitations that should be addressed. First, the data were exclusively self-reported, and we did not obtain additional information regarding mental health diagnoses or treatment history. Second, although participants recruited from the community are more representative compared with those recruited from clinical units, the volunteers may have had various motivations for participating in the study. Third, we did not assess the content of the Internet activity nor psychiatric diagnoses. Fourth, we did not survey factors that may relate to change in IA in college students, for example, personal traits, socialization process, and enthusiasm devoted to study. Lastly, the four subgroups (PNIA, NIA, RIA, and PIA) were of unequal size; two were comparably small (*n* = 20 in NIA; *n* = 30 in PIA). The small sample size might limit the power of statistical analysis.

## 5. Conclusions

The results of this study indicated that impulse control difficulties predicted the incidence of IA during the follow-up period of 1 year among male college students, whereas IA did not significantly predict the change in difficulties in emotion regulation. Mental health professionals should help college students develop effective strategies for emotion regulation.

## Figures and Tables

**Figure 1 ijerph-17-04766-f001:**
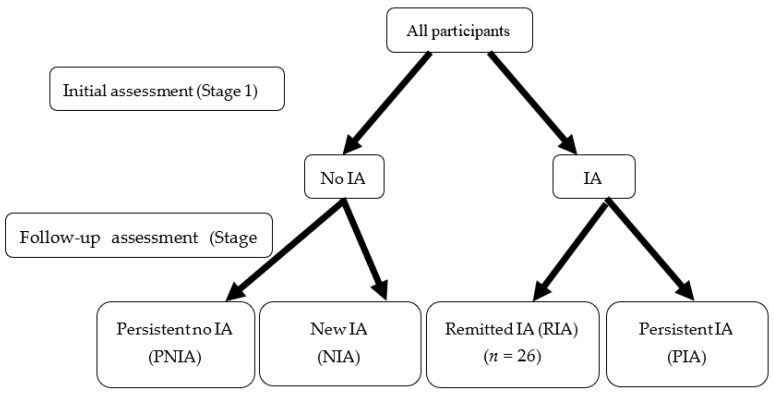
Study participants were assigned to one of four addiction groups. IA: Internet addiction

**Table 1 ijerph-17-04766-t001:** Comparisons of demographic data and difficulties in emotion regulation between participants who participated in and who did not participate in follow-up assessment.

	No Follow-Up (*n* =176) *n* (%) or mean (SD)	Follow-Up (*n* = 324) *n* (%) or mean (SD)	χ^2^ or *t*
Gender, *n* (%)			
Female	93 (52.8)	169 (52.2)	0.884
Male	83 (47.2)	155 (47.8)	
Age (years), mean (SD)	21.9 (1.5)	22.3 (1.9)	0.047
CIAS, mean (SD)	56.6 (13.2)	55.0 (14.2)	1.189
DERS, mean (SD)			
Non-acceptance of emotional responses	12.7 (4.8)	12.6 (3.9)	0.273
Difficulties engaging in goal-directed behavior	12.2 (3.9)	12.0 (3.6)	0.539
Impulse control difficulties	11.2 (4.0)	11.5 (4.2)	−0.802
Lack of emotional awareness	15.3 (4.5)	14.7 (3.9)	1.364
Limited access to emotion regulation strategies	17.1 (5.7)	16.7 (5.3)	0.819
Lack of emotional clarity	9.9 (3.4)	10.0 (2.9)	−0.350

CIAS: Chen Internet Addiction Scale; DERS: Difficulties in Emotion Regulation Scale; SD: Standard deviation.

**Table 2 ijerph-17-04766-t002:** Comparisons of demographic data and difficulties in emotion regulation between the PNIA and NIA groups and the RIA and PIA groups.

	PNIA Group (*n* = 248) *n* (%) or mean (SD)	NIA Group (*n* = 20) *n* (%) or mean (SD)	χ^2^ or *t*	RIA Group (*n* = 26) *n* (%) or mean (SD)	PIA Group (*n* = 30) *n* (%) or mean (SD)	χ^2^ or *t*
Gender, *n* (%)						
Female	129 (50.0)	11 (55)	0.066	15 (57.7)	14 (46.7)	0.678
Male	119 (48.0)	9 (45)		11 (42.3)	16 (53.3)	
Age (years), mean (SD)	22.3 (2.0)	22.4 (1.9)	−0.347	21.6 (1.2)	22.6 (1.9)	−2.421
DERS, mean (SD)						
Non-acceptance of emotional responses	12.2 (3.9)	13.6 (3.2)	−1.532	13.7 (3.6)	14.3 (4.7)	−0.601
Difficulties engaging in goal-directed behavior	11.4 (3.3)	11.9 (3.1)	−0.605	14.8 (3.4)	14.8 (4.3)	−0.029
Impulse control difficulties	10.8 (3.8)	13.4 (4.1)	−2.902 *	14.0 (4.7)	14.2 (5.0)	−0.179
Lack of emotional awareness	14.5 (4.1)	14.0 (2.8)	0.514	16.3 (3.4)	16.0 (3.1)	0.312
Limited access to emotion regulation strategies	15.7 (4.8)	18.5 (4.2)	−2.481	20.1 (6.0)	20.4 (5.6)	−0.229
Lack of emotional clarity	9.7 (2.9)	10.5 (2.9)	−1.168	11.0 (2.1)	11.0 (2.7)	−0.052

* *p* < 0.008. DERS: Difficulties in Emotion Regulation Scale; NIA: No Internet addiction; PIA: Persistent Internet addiction; PNIA: Persistent no Internet addiction; RIA: Remitted Internet addiction; SD: Standard deviation.

**Table 3 ijerph-17-04766-t003:** Interactions between Internet addiction and changes in difficulty in emotion regulation analyzed by repeated-measures ANOVA.

	Within-Subject Analysis					Between-Subject Analysis			
	df	Mean Square	F	*p*		df	Mean Square	F	*p*
Non-acceptance of emotional responses									
Cohort	1	0.28	0.03	0.859	Intercept	1	64,390.83	2769.62	<0.001
Cohort × Internet addiction	1	10.75	1.22	0.271	Internet addiction	1	178.02	7.66	0.006
	322	8.84			Error	322	23.25		
Difficulties engaging in goal-directed behavior									
Cohort	1	0.61	0.10	0.755	Intercept	1	63,252.18	3477.03	<0.001
Cohort × Internet addiction	1	12.34	1.96	0.162	Internet addiction	1	831.19	45.69	<0.001
Error	322	6.29			Error	322	18.19		
Impulse control difficulties									
Cohort	1	45.45	5.81	0.016	Intercept	1	55,047.18	2220.87	<0.001
Cohort × Internet addiction	1	24.46	3.13	0.078	Internet addiction	1	655.32	26.44	<0.001
Error	322	7.82			Error	322	24.79		
Lack of emotional awareness									
Cohort	1	331.13	46.49	<0.001	Intercept	1	76,173.83	3796.15	<0.001
Cohort × Internet addiction	1	40.72	5.72	0.017	Internet addiction	1	96.71	4.82	0.029
Error	322	7.12			Error	322	20.07		
Limited access to emotion regulation strategies									
Cohort	1	44.38	3.87	0.050	Intercept	1	116,832.71	3059.68	<0.001
Cohort × Internet addiction	1	60.46	5.28	0.022	Internet addiction	1	1150.13	30.12	<0.001
Error	322	11.46			Error	322	38.19		
Lack of emotional clarity									
Cohort	1	41.63	10.91	0.001	Intercept	1	37,474.75	3280.47	<0.001
Cohort × Internet addiction	1	0.07	0.02	0.892	Internet addiction	1	151.88	13.30	<0.001
Error	322	3.82			Error	322	11.42		

**Table 4 ijerph-17-04766-t004:** Comparisons of difficulties in emotion regulation between the initial (Stage 1) and follow-up (Stage 2) investigations among the participants with and without Internet addiction at Stage 1.

	Stage 1 Mean (SD)	Stage 2 Mean (SD)	Paired *t*	*p*
Internet addiction group (*n* = 56)				
Non-acceptance of emotional responses	14.0 (4.2)	13.7 (4.2)	0.582	0.563
Difficulties engaging in goal-directed behavior	14.8 (3.9)	14.3 (3.8)	0.787	0.435
Impulse control difficulties	14.1 (4.8)	12.9 (4.3)	1.931	0.059
Lack of emotional awareness	16.1 (3.2)	13.6 (3.4)	5.500	<0.001
Limited access to emotion regulation strategies	20.3 (5.7)	18.8 (5.3)	2.084	0.042
Lack of emotional clarity	11.0 (2.4)	10.4 (3.1)	1.520	0.134
Non-Internet addiction group (*n* = 268)				
Non-acceptance of emotional responses	12.3 (3.8)	12.7 (4.1)	−1.504	0.134
Difficulties engaging in goal-directed behavior	11.4 (3.3)	11.7 (3.6)	−1.372	0.171
Impulse control difficulties	11.0 (3.9)	10.8 (3.9)	0.808	0.420
Lack of emotional awareness	14.4 (4.0)	13.2 (3.5)	5.243	<0.001
Limited access to emotion regulation strategies	16.0 (4.8)	16.1 (4.9)	−0.407	0.684
Lack of emotional clarity	9.8 (2.9)	9.1 (2.6)	4.275	<0.001

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
