# Peer review of "Relationship between Difficulty in Emotion Regulation and Internet Addiction in College Students: A One-Year Prospective Study"

_ijerph, 2020, doi:10.3390/ijerph17134766_

Round 1
Reviewer 1 Report
Dear Authors,
Thank you very much for enabling me to read your manuscript. The topic is really significant nowadays when all society is online. My suggestions and questions for a possible improvement:
- Were there any gender differences?
- Which intervention strategies could be used? Please describe them in the part on Discussion.
Thanks.
Reviewer
Author Response
Comment 1
Were there any gender differences?
Response
Thank you for your comment. We found that the gender difference did exist in the prediction of impulse control difficulties for the incidence of IA. We added the new result into Results and Discussion sections of the revised manuscript as below.
“We further stratified the participants by gender and found that the prediction of impulse control difficulties for the incidence of IA existed in only male (t = -2.875, p = 0.005) but not in female participants (t = -1.270, p = 0.206). We further stratified the participants by gender and found that the prediction of impulse control difficulties for the incidence of IA existed in only male (t = -2.875, p = 0.005) but not in female participants (t = -1.270, p = 0.206).” (Please refer to line 168-170)
“The present study found the gender difference in the prediction of impulse control difficulties for the incidence of IA. Previous studies have also found gender differences in the psychopathology of IA [45-46]. However, the numbers of participants with the incidence of IA during the one-year follow-up period were small. Further study with more participants are warranted to replicate this result.” (Please refer to line 219-223)
Comment 2
Which intervention strategies could be used? Please describe them in the part on Discussion.
Response
We added intervention strategies for impulse control difficulties for IA into Discussion section as below. Please refer to line 216-219.
“Cognitive-behavioral therapy for IA aims to increase the clients’ ability of impulse control by training clients’ skills to monitor their inner feelings and control impulse behaviors [41-43]. Research also found that electro-acupuncture had an advantage over psychological intervention in terms of impulsivity control in adolescents with IA [44].”
Reviewer 2 Report
This study aimed to examine whether the relationships between IA and emotion regulation vary according to various dimensions of emotion regulation in a sample of college students. The research question is important and timely, but I have several comments.
Abstract: Please present important quantitative findings (with statistical significance) in abstract.
Line 52: Need citations for this – “College students may use the Internet for studying, gaming, social networking, gambling, chatting, shopping, and watching pornographic videos”.
Line 56: With the word “indicating”, do you mean that depression and social anxiety are associated with or lead to difficulties in emotion regulation? If so, provide citation. Need to elaborate more. “A review study reported that IA was positively associated with depression and social anxiety [11], indicating that people with IA may have difficulties in emotion regulation [12].”
Line 147: Over one thirds of participants dropped out from stage-1 to stage-2 is. What are the reasons for high dropout rate? Please provide a demographic comparison for stage-2 completers vs non-completers. Also, provide stage-1 survey data for those 2 groups.
Line 173: Authors say that “influence of IA at Stage 1 on changes of DERS scores from Stage 1 to Stage 2 are shown in Table 3”. However, according to your hypothesis, aren’t you testing the opposite direction (whether IA influences DERS scores)?
Line 174: Authors say that “The results indicated that scores for the subscales of lack of emotional awareness (p = 0.017) and limited access to emotion regulation strategies (p = 0.022) tended to decrease more from Stage 1 to Stage 2 in the IA group than in the non-IA group; however, the results did not reached the statistically significant level.” Please elaborate this more. Alternatively, you would better use between-group analysis to present your main findings.
Line 178: Your main finding is that more severe impulse control difficulties predicted a higher incidence of IA during the follow-up period of 1 year, but your results section does not elaborate it. Please summarize the between-group analysis findings from Table-3.
Author Response
Comment 1
Abstract: Please present important quantitative findings (with statistical significance) in abstract.
Response
We added the values of t and p of t test (t = -2.875, p = 0.005) for the prediction of impulse control difficulties for the incidence of IA into Abstract of the revised manuscript. Please refer to line 32.
Comment 2
Line 52: Need citations for this – “College students may use the Internet for studying, gaming, social networking, gambling, chatting, shopping, and watching pornographic videos”.
Response
We added two citations for this sentence as below. Please refer to line 54 and 289-294.
- Anderson, K.J. Internet use among college students: An exploratory study, Am. Coll. Health 2001, 50, 1, 21-26. dio: 10.1080/07448480109595707
- Pal Singh Balhara, Y.; Doric, A.; Stevanovic, D.;Knez, ; Swarndeep ingh, S.; Chowdhury, M.R.R;, Kafali, H.Y.; Sharma, P.; Vally, Z.; Vu, T.V.; et al. Correlates of problematic Internet use among college and university students in eight countries: An international cross-sectional study. Asian J. Psychiatr. 2019, 45, 113-120. doi:10.1016/j.ajp.2019.09.004
Comment 3
Line 56: With the word “indicating”, do you mean that depression and social anxiety are associated with or lead to difficulties in emotion regulation? If so, provide citation. Need to elaborate more. “A review study reported that IA was positively associated with depression and social anxiety [11], indicating that people with IA may have difficulties in emotion regulation [12].”
Response
Thank you for your comment. We revised this paragraph as below. Please refer to line 60-68.
“Emotion regulation is the attempts to alter emotional experiences via the initiation, maintenance or modification of frequency, intensity or duration of emotional experiences [19]. Difficulties with emotion regulation are believed to be risk factors for addiction [20,21]. Gratz and Roemer proposed that emotion regulation involves multiple conceptions including (a) awareness and understanding of emotions, (b) acceptance of emotions, (c) ability to control impulsive behaviors and behave in accordance with desired goals when experiencing negative emotions, and (d) ability to use situationally appropriate emotion regulation strategies flexibly to modulate emotional responses as desired in order to meet individual goals and situational demands [22]. The relative absence of any or all of these abilities would indicate the presence of difficulties in emotion regulation [22].”
Comment 4
Line 147: Over one thirds of participants dropped out from stage-1 to stage-2 is. What are the reasons for high dropout rate? Please provide a demographic comparison for stage-2 completers vs non-completers. Also, provide stage-1 survey data for those 2 groups.
Response
We added the results of comparing demographic data and the levels of IA on the CIAS and difficulties in emotion regulation on the DERS at initial assessment between these two groups into Results section as below and Table 1. Please refer to line 152-159.
“Of 176 participants (93 women and 83 men) who did not participate in follow-up assessment, 96 (54.5%) were could not be contacted, 48 (27.3%) refused to participate in follow-up assessment, and 32 (18.2%) had the desire to participate but were unable to do so due to work or army service. The results of comparing demographic data and the levels of IA on the CIAS and difficulties in emotion regulation on the DERS at initial assessment between participants who participate in and who did not participate in follow-up assessment are shown in Table 1. No differences were found in gender, age, and the levels of IA and difficulties in emotion regulation between these two groups (all p > 0.008).”
Comment 5
Line 173: Authors say that “influence of IA at Stage 1 on changes of DERS scores from Stage 1 to Stage 2 are shown in Table 3”. However, according to your hypothesis, aren’t you testing the opposite direction (whether IA influences DERS scores)?
Response
Thank you for your comment. We really tested whether IA influences DERS scores. Therefore, we revised this sentence into “…the effect of IA at Stage 1 on changes of DERS scores from Stage 1 to Stage 2 are shown in Table 3”. Please refer to line 186.
Comment 6
Line 174: Authors say that “The results indicated that scores for the subscales of lack of emotional awareness (p = 0.017) and limited access to emotion regulation strategies (p = 0.022) tended to decrease more from Stage 1 to Stage 2 in the IA group than in the non-IA group; however, the results did not reached the statistically significant level.” Please elaborate this more. Alternatively, you would better use between-group analysis to present your main findings.
Response
Thank you for your reminding. In the revised manuscript we added the results of between-group analysis as below. Please refer to line 191-200.
“Regarding the results of between-subject analysis, there were significant differences in non-acceptance of emotional responses (p = 0.006), difficulties engaging in goal-directed behavior (p < 0.001), impulse control difficulties (p < 0.001), limited access to emotion regulation strategies (p < 0.001), and lack of emotional clarity (p < 0.001) between the participants with and without IA at Stage 1.
To explore the effects of IA on the changes of difficulties in emotional regulation, the scores on the DERS were further compared between the Stage 1 and Stage 2 investigations using paired t-test in the participants with and without IA at Stage 1 (Table 4). The results of paired t-test demonstrated that in the IA group, the scores on the subscale of lack of emotional awareness decreased significantly from Stage 1 to Stage 2. In the non-IA group, the scores on the subscales of lack of emotional awareness and lack of emotional clarity decreased significantly from Stage 1 to Stage 2.”
Comment 7
Line 178: Your main finding is that more severe impulse control difficulties predicted a higher incidence of IA during the follow-up period of 1 year, but your results section does not elaborate it. Please summarize the between-group analysis findings from Table-3.
Response
- We elaborated the main finding in Results section as below. Please refer to line 167-168. We also discuss it in the first paragraph of Discussion section. Please refer to line 204-223.
“…impulse control difficulties at Stage 1 predicted the incidence of IA at Stage 2 during the follow-up period of 1 year.”
- We added the results of between-group analysis into the revised manuscript. Please refer to line 191-200.
Reviewer 3 Report
The merit of this work is that a real longitudinal study is prresented with an interval of a year between two measurement times for two assessments of internet addiction and various facets of emotion regulation. Thus, implications on causal effects can be drawn between these two measures on a sample of initially n = 500 participants.
Too often we read about cross-sectional studies which make unwarranted claims about causal mechanisms. This positive aspect could be highlighted to a stronger degree. Surrounding these "causal implications" one could make the design more clear at the introduction at lines 76ff. The possible causal directions (however, often the term "relationship" is used which is appropriate already for cross-sectional studies) are given in both directions as internet addiction => emotion regulation (lines 76-77), emotion regulation => internet addiction (lines 79-81), and again internet addiction => emotion regulation (lines 81-82). In these sentences the causal hypotheses go in both directions without stating how exactly the design of the study might lead to possible answers of the directionality of results. This would be my main point, to make the underlying logic more transparant.
Here a few minor points:
- Line 104: Why is there a two-number cut-off point 67/68 for the range of possible scores between 26 and 104? Shouldn't there be just one number?
- Education was not assessed. Education could be an important mediator/ moderator of internet addiction and emotion regulation.
- I doubt that impulse control difficulties (lines186ff) alone affect social bonds (and would be a reason to go onto the internet instead). Impulsive people find each other or can even be attractive (fun) for more self-controlled people. Other personality traits might be more strongly related to internet use and abuse (do the authors have ideas / references)?
- In general the four subgroups (PNIA, NIA, RIA, PIA) are of unequal size and three are comparably small (between n = 20 and 30). I understand that the initial size of 500 is large and having n = 324 students at t2 is not bad. For pragmatic reasons one has to deal with these outcomes. One could discuss this as a limitation.
Author Response
Comment 1
Too often we read about cross-sectional studies which make unwarranted claims about causal mechanisms. This positive aspect could be highlighted to a stronger degree. Surrounding these "causal implications" one could make the design more clear at the introduction at lines 76ff. The possible causal directions (however, often the term "relationship" is used which is appropriate already for cross-sectional studies) are given in both directions as internet addiction => emotion regulation (lines 76-77), emotion regulation => internet addiction (lines 79-81), and again internet addiction => emotion regulation (lines 81-82). In these sentences the causal hypotheses go in both directions without stating how exactly the design of the study might lead to possible answers of the directionality of results. This would be my main point, to make the underlying logic more transparant.
Response
Thank you for your suggestion. We revised this paragraph as below. Please refer to line 82-87.
“No prospective study has examined the bidirectional relationship between IA and difficulties in emotion regulation. The prospective study design can provide the temporal relationships between IA and difficulties in emotion regulation and infer the causal relationship between them. The aims of this prospective study were to evaluate the predictive effect of difficulty in emotion regulation on the occurrence and remission of IA, and to determine whether IA can predict the change of difficulty in emotion regulation among college students during the follow-up period of 1 year.”
Comment 2
Line 104: Why is there a two-number cut-off point 67/68 for the range of possible scores between 26 and 104? Shouldn't there be just one number?
Response
We revised it into “…the 68 of the total CIAS score has the highest diagnostic accuracy and accepted sensitivity and specificity for IA [26].” Please refer to line 110.
Comment 3
Education was not assessed. Education could be an important mediator/ moderator of internet addiction and emotion regulation.
Response
Because that all participants were college students, we did not assess participants; education level.
Comment 4
I doubt that impulse control difficulties (lines186ff) alone affect social bonds (and would be a reason to go onto the internet instead). Impulsive people find each other or can even be attractive (fun) for more self-controlled people. Other personality traits might be more strongly related to internet use and abuse (do the authors have ideas / references)?
Response
Thank you for your suggestion. We added new contents regarding the personality traits that may relate to Internet addiction as below. Please refer to line 57-68.
“There have many studies examining the relationship between IA and personality characteristics such as neuroticism [13-15], conscientiousness [13-15], and agreeableness [13-15], boredom proneness [16], borderline personality characteristics [17], reinforcement sensitivity [18] and frustration intolerance [18]. Emotion regulation is the attempts to alter emotional experiences via the initiation, maintenance or modification of frequency, intensity or duration of emotional experiences [19]. Difficulties with emotion regulation are believed to be risk factors for addiction [20,21]. Gratz and Roemer proposed that emotion regulation involves multiple conceptions including (a) awareness and understanding of emotions, (b) acceptance of emotions, (c) ability to control impulsive behaviors and behave in accordance with desired goals when experiencing negative emotions, and (d) ability to use situationally appropriate emotion regulation strategies flexibly to modulate emotional responses as desired in order to meet individual goals and situational demands [22]. The relative absence of any or all of these abilities would indicate the presence of difficulties in emotion regulation [22].”
Comment 5
In general the four subgroups (PNIA, NIA, RIA, PIA) are of unequal size and three are comparably small (between n = 20 and 30). I understand that the initial size of 500 is large and having n = 324 students at t2 is not bad. For pragmatic reasons one has to deal with these outcomes. One could discuss this as a limitation.
Response
Thank you for your reminding. We listed it as one of limitations of this study as below. Please refer to line 256-258.
“The four subgroups (PNIA, NIA, RIA and PIA) were of unequal size; three were comparably small (n = 20 in NIA; n = 30 in PIA). The small sample size might limit the power of statistical analysis.”
Round 2
Reviewer 2 Report
Thank you addressing my comments. I have no further comments.